# LEARNING REPRESENTATIONS ON LP HYPERSPHERES: THE EQUIVALENCE OF LOSS FUNCTIONS IN A MAP APPROACH

## ABSTRACT

A common practice when training Deep Neural Networks is to force the learned representations to lie on the standard unit hypersphere, with respect to the $L_2$ norms. Such practice has been shown to improve both the stability and final performances of DNNs in many applications. In this paper, we derive a unified theoretical framework for learning representation on any $L_p$ hyperspheres for classification tasks, based on Maximum A Posteriori (MAP) modeling. Specifically, we give an expression of the probability distribution of multivariate Gaussians projected on any $L_p$ hypersphere and derive the general associated loss function. Additionally, we show that this framework demonstrates the theoretical equivalence of all projections on $L_p$ hyperspheres through the MAP modeling. It also provides a new interpretation of traditional Softmax Cross Entropy with temperature (SCE-$\tau$) loss functions. Experiments on standard computer vision datasets give an empirical validation of the equivalence of projections on $L_p$ unit hyperspheres when using adequate objectives. It also shows that the SCE-$\tau$ on projected representations, with optimally chosen temperature, shows comparable performances. The code is publicly available at `https://anonymous.4open.science/r/map_code-71C7/`.

## 1 INTRODUCTION

Cross-entropy (CE) is the most commonly used loss function for classification, even though it is often modified Ahn et al. (2021); Caccia et al. (2022); Wang et al. (2017) or coupled with additional loss terms Hinton et al. (2015); Li et al. (2019). On the other hand, many studies in the literature address designing output normalization. Bouchard (2007) introduced upper bounds for improving softmax computation stability. De Brebisson & Vincent (2015) introduced a family of functions behaving as normalizing functions and gave experimental justifications for softmax alternatives. Other sparse alternatives have similarly been developed Martins & Astudillo (2016); Laha et al. (2018); Liu et al. (2017). Further studies considered the probabilistic modeling of the trained feature space explicitly. Wan et al. (2018) have leveraged a Gaussian mixture model coupled with a CE. Additional studies also consider a similar setting, opposing the obtained loss function to the traditional Softmax Cross Entropy (SCE) Yan et al. (2020).

It is known that the softmax operation can be interpreted as resulting from the formulation of the a posteriori distribution of the class given the data and that the search for the a posteriori maximum leads, with a Gaussian assumption, to the standard cross-entropy criterion; see for example (Bishop, 2006, Section 4.2, pages 197-199).

A standard practice when training Deep Neural Networks is to force the learned representations to lie on the standard unit hypersphere, with respect to the $L_2$ norms. Such practice has been shown to improve both the stability and final performances of DNNs in many applications, see e.g. Wang et al. (2017); Tian et al. (2019); Zimmermann et al. (2021); Chen et al. (2020). However, this is usually not directly accounted for when deriving a loss function for the whole classification process, including the projection step.

In this paper, we first recall the MAP approach for the DNN classification problem and give an explicit connection to SCE and its variant Softmax Cross-Entropy with temperature (SCE-$\tau$) and

bias Zhang et al. (2018); Agarwala et al. (2020). Indeed, we show that SCE can be interpreted as a MAP with a class-conditional isotropic Gaussian hypothesis on the standard *scaled*-simplex (the standard simplex scaled by a factor $r$). Similarly, we demonstrate that the temperature parameter used for re-scaling the network outputs in SCE-$\tau$ can be expressed as the ratio between the scaling factor $r$ and the Gaussian distributions $v$ variance. The insights given by the MAP approach allow us to give a meaningful interpretation of the SCE and, more than that, to consider more general scenarios. Specifically, we investigate the impact of a particular family of nonlinear output transformations: projections onto $L_p$ hyperspheres, notably to compare performances with SCE and assess the impact of $p$. While the already mentioned $L_2$ projections are commonly adopted, the general case of $L_p$ projections is widely unexplored. Different $L_p$ norms change the geometry of hyperspheres, affecting how data is projected and separated. For instance, with $p > 2$, hyperspheres become more flattened, while $p < 2$ makes them more angular, which can enhance class separation in certain directions, see Figure 1.

Building on the MAP approach to learn representations, we derive the expression of the probability distribution for Gaussian distributions projected on general $L_p$ hyperspheres. This introduces the Projected Gaussian Distribution (PGD), a generalization of the Angular Gaussian Distribution presented in Michel et al. (2024). From this expression, we establish the theoretical equivalence of all $L_p$ projections in the MAP setting. Eventually, we experiment with PGD through the MAP framework as well as with SCE-$\tau$ on output projected on the $L_p$ unit-sphere. Finally, we conclude that PGD and SCE-$\tau$ can lead to **comparable performances, in case of a $L_p$ projection layer**, provided optimal $v$ values are used for any values of $p$ and show that leveraging PGD or projecting on the hypercube can **improve stability** concerning the variance. In summary, we make the following contributions:

- we highlight a connection between the MAP approach and SCE variants, which give additional insight on the loss function;

- we propose an expression of PGD, the distribution of a Gaussian distribution on any $L_p$ hypersphere;

- we show that projecting on the hypercube or leveraging PGD benefits stability with regard to $v$, while maintaining performance on par with the best SCE-$\tau$ strategy.

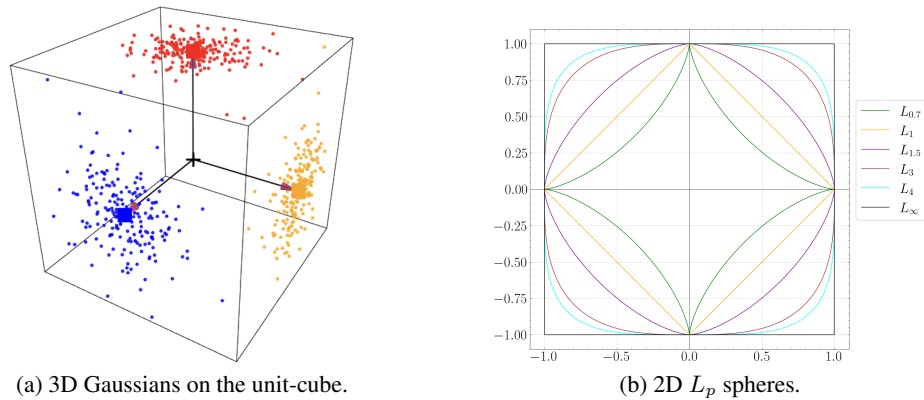

(a) 3D Gaussians on the unit-cube.    (b) 2D $L_p$ spheres.

Figure 1: (a) Illustration of Gaussian-sampled points projected onto 3D unit-cube. Gaussians are centered around the standard basis. (b) 2D $L_p$ hyperspheres visualisation for various $p$ values.

## 2 RELATED WORK

In this section, we give a short overview of related works and concepts.

**Softmax-Cross Entropy and its variants.** One of the most widely used loss functions for classification tasks is the Cross-Entropy, commonly combined with the softmax function applied to the output layer Goodfellow et al. (2016). Numerous works have been proposed as alternatives to the traditional softmax operator, such as sparse alternative Martins & Astudillo (2016); Liu et al. (2017);

Laha et al. (2018) or spherical softmax De Brebisson & Vincent (2015). Similarly, prototype-based alternatives to SCE have been developed Bytyqi et al. (2023); Wei et al. (2023); Mettes et al. (2019). Another variant of SCE introduces a temperature parameter Wang et al. (2017); Pang et al. (2019), which results in the following loss function:

$$\mathcal{L}_{CE}(\boldsymbol{z}) = -\sum_{c=1}^{L} \mathbb{1}(y = c) \log \frac{e^{z_c/\tau}}{\sum_{j=1}^{L} e^{z_j/\tau}} \tag{1}$$

where $y$ is the true label of $\boldsymbol{z}$, $L$ the total number of classes, $z_c$ the $c^{th}$ component of $\boldsymbol{z}$ and $\tau \in \mathbb{R}^{+\star}$ the temperature. The usage of temperature similarly goes beyond SCE and has been studied in contrastive learning Zhang et al. (2021); Khosla et al. (2020); Chen et al. (2020). However, such studies are mostly empirical, and there is a lack of studies going beyond intuition.

**MAP for learning representations.** Maximum A Posteriori is a fundamental probabilistic method and has been applied to countless problems Gauvain & Lee (1994); Santini & Del Bimbo (1995); in the context of DNNs, Michel et al. Michel et al. (2024) applied a natural MAP framework for learning representations on the unit-hypersphere. Other probabilistic modeling also derived similar loss functions Hasnat et al. (2017). However, to the best of our knowledge, no explicit link to the SCE-$\tau$ loss and its implication in terms of interpretation has been developed in earlier research.

**Projection on $L_p$ hyperspheres.** While projection on the unit-hypersphere (a.k.a. normalization) is a common practice in representation learning Grill et al. (2020); Khosla et al. (2020); Mettes et al. (2019); Michel et al. (2024), it is often bounded to the $L_2$ hypersphere. In adversarial training, $L_\infty$ metric is also used for measuring the distance between the original and the attacked sample Mao et al. (2019); Tramer & Boneh (2019). Few studies of the general family of $L_p$ projections for DNNs in the context of image classification exist. An attempt to leverage $L_p$ normalization of the penultimate layer can be found in Trivedi et al. (2022).

## 3 FROM MAP TO SCE

In this section, we recall that SCE, and even SCE-$\tau$ can be recovered as special cases from the MAP learning framework. This provides the groundwork for presenting further extensions of cost functions, incorporating the notion of projection on $L_p$ hyperspheres.

### 3.1 POSTERIOR EXPRESSION FROM LATENT REPRESENTATION

As introduced in Section 1, we are interested in expressing the posterior $p(c|x)$. We start from the consideration that Deep Neural Networks (DNN) are fundamentally encoders that can learn a mapping between an input $x \in \mathbb{R}^D$ to a latent representation $\boldsymbol{z} \in \mathbb{R}^d$, where $D$ and $d$ are the dimensions of the input and the latent representation respectively and $D \gg d$. From this point, the posterior estimation problem becomes estimating $p(c|\boldsymbol{z})$. By the Bayes rule and expressing $f(\boldsymbol{z})$ by marginalizing across considered classes, $p(c|\boldsymbol{z})$ can be expressed as:

$$p(c|\mathbf{z}) = \frac{f_c(\mathbf{z})\pi_c}{\sum_{\ell=1}^{L} f_\ell(\mathbf{z})\pi_\ell} \tag{2}$$

where $\pi_\ell$ are class priors, $L$ the total number of classes and $f_\ell(z)$ the conditional p.d.f. of $\boldsymbol{z}$ given $c$. Considering latent representations as DNN outputs such that $\boldsymbol{z} = \Phi_\theta(x)$, with $\theta$ the trainable DNN parameters; we can rewrite previous expression:

$$p(c|\mathbf{z}) = \frac{f_c(\Phi_\theta(x))\pi_c}{\sum_{\ell=1}^{L} f_\ell(\Phi_\theta(x))\pi_\ell} \tag{3}$$

### 3.2 MAXIMUM A POSTERIORI LOG-LOSS

For a set of $b$ of observations $(\mathbf{z}_i, y_i)_{1 \le i \le b}$, where the $y_i \in [\![1, L]\!]$ are the labels of classes and $\boldsymbol{z}_i \in \mathbb{R}^d$, we want to maximize $p(y_1 \cdots y_b | \mathbf{z}_1 \cdots \mathbf{z}_b)$. Let us consider such observations to be independent. The

objective becomes maximizing $\prod_{c=1}^{L}\prod_{i\in I_c}p(c|\mathbf{z}_i)$ with $I_c = \{i \in [\![1,b]\!] \mid y_i = c\}$. The posterior distribution can thus be expressed as

$$p(y_1 \cdots y_b | \mathbf{z}_1 \cdots \mathbf{z}_b) = \prod_{c=1}^{L}\prod_{i\in I_c}\frac{f_c(\mathbf{z}_i)\pi_c}{\sum_{\ell=1}^{L}f_\ell(\mathbf{z}_i)\pi_\ell}. \tag{4}$$

A more practical log-loss form can obtained from equation 4 by taking the average of the logarithm:

$$\mathcal{L}_{\mathrm{MAP}}^{\mathrm{log}}(\mathcal{B},\theta) = -\frac{1}{|\mathcal{B}|}\sum_{c=1}^{L}\sum_{i\in I_c}\log\frac{f_c\big(\Phi_\theta(\mathbf{x}_i)\big)\pi_c}{\displaystyle\sum_{\ell=1}^{L}f_l\big(\Phi_\theta(\mathbf{x}_i)\big)\pi_\ell} \tag{5}$$

With $|\mathcal{B}|$ the size of batch $\mathcal{B} = (x_i, y_i)_{1\leq i \leq b}$. In the MAP framework, we minimize $\mathcal{L}_{MAP}^{log}$ as described in equation 5.

### 3.3 GAUSSIAN HYPOTHESIS AND EQUAL PRIORS

The MAP framework described above heavily depends on the choice of the class-conditional p.d.f. $f_c(.)$. A reasonable assumption is that these p.d.f follow a Gaussian distribution and that all priors are equal. Thus, we derive Proposition 3.1 and give the proof in Appendix A.

**Proposition 3.1.** *Let* $\{r_l\}_{1\leq l \leq L}$ *be a basis of* $\mathbb{R}^L$ *such that* $r_l = r \cdot e_l$ *with* $r \in \mathbb{R}$ *and* $\{e_l\}_{1\leq l \leq L}$ *the standard basis of* $\mathbb{R}^L$. *Under the following assumptions:*

- *the conditional probability density functions* $\{f_l(.)\}_{1\leq l \leq L}$ *follow an isotropic Gaussian distribution of variance* $v$ *centered around means* $\{r_l\}_{1\leq l \leq L}$;

- *classes priors* $\{\pi_l\}_{1\leq l \leq L}$ *are equal;*

*the loss* $\mathcal{L}_{\mathrm{MAP}}^{\mathrm{log}}$ *takes the following form:*

$$\mathcal{L}_{MAP}^{log}(\mathcal{B},\theta) = -\frac{1}{|\mathcal{B}|}\sum_{c=1}^{L}\sum_{i\in I_c}\log\frac{e^{\frac{r}{v}\Phi_\theta(\boldsymbol{x}_i)_c}}{\displaystyle\sum_{\ell=1}^{L}e^{\frac{r}{v}\Phi_\theta(\boldsymbol{x}_i)_l}} \tag{6}$$

*with* $\Phi_\theta(\boldsymbol{x}_i)_c$ *the c-th component of* $\Phi_\theta(\boldsymbol{x}_i)$, *the output of the model given the input* $x_i$.

### 3.4 CONNEXION WITH SCE AND ITS VARIANTS

Under simple assumptions, the MAP framework leads to the $\mathcal{L}_{MAP}^{log}$ as defined in equation 6. When $\frac{r}{v} = 1$, we recover the usual SCE loss. Additionally, if we define $\tau = \frac{v}{r}$, then we recover the SCE-$\tau$ loss. Thus, we can interpret SCE-$\tau$ as a MAP with a class conditional Gaussian hypothesis on the standard *scaled*-simplex whose scaling ratio $r$ and variance $v$ are conditioned such that $\frac{r}{v} = \tau$. This statement similarly holds for SCE when $\tau = 1$. Furthermore, the Softmax operation appears naturally in this modeling. From this interpretation, two scenarios can be identified. If the learned representation is projected on the unit-hypersphere, $r = 1$, and if we assume that these projections are also Gaussian, then variance $v$ follows by the remodeling as $\tau = v$.

Of course, the Gaussian assumption of the projection on the hypersphere is questionable. In Section 4.4, we discuss the validity of this assumption and we give the expression of the Projected Gaussian Distribution in Section 4. Moreover, if the learned representations are not constrained, it follows that $r$ and $v$ are learned such that the relation $\tau = \frac{v}{r}$ is respected.

Another popular practice when tackling classification problems is prototype learning Zhang et al. (2020); Lin et al. (2023); Yang et al. (2018); Ho et al. (2021); Wei et al. (2023); De Lange & Tuytelaars (2021). The main idea is to compare the learned representations to a set of prototypes $\mathcal{P} = \{\boldsymbol{p_1}, \cdots, \boldsymbol{p_L}\}$. The probabilities are computed using a modified version of the softmax, such as detailed in Equation 7.

$$\mathrm{ProtoSoftmax}(\mathbf{z}, P)_i = \frac{e^{\boldsymbol{z}\cdot\boldsymbol{p_i}}}{\sum_{j=1}^{L}e^{\boldsymbol{z}\cdot\boldsymbol{p_j}}} \tag{7}$$

Moreover, several studies introduce an additional class-dependent coefficient in the softmax operator, referred to as softmax with bias or re-weighted softmax: Jodelet et al. (2021); Ren et al. (2020); Legate et al. (2023).

**Proposition 3.2.** *Starting from the MAP log-loss defined in Equation 5, under the following assumptions:*

- *The prototypes $\mathcal{P}$ lie on a hypersphere.*

- *The conditional probability density functions $\{f_l(.)\}_{1 \leq l \leq L}$ follow an isotropic gaussian distribution of variance $v$ centered around means $\mathcal{P}$*

- *The variance $v$ of the isotropic Gaussians is equal to one.*

*Then, the MAP log-loss is equivalent to the SCE with prototype and bias loss.*

# 4 LEARNING ON THE $L_p$ HYPERSPHERE

We showed that minimizing an SCE-$\tau$ objective with representations learned on the unit-sphere gives control over the Gaussian variance, provided that the projection itself is considered Gaussian. In this section, we discuss the impact of invertible and non-invertible transformations on the resulting distribution and on MAP training objective, in the case of projections on $L_p$ hyperspheres.

## 4.1 MAP WITH ADDITIONAL TRANSFORMATIONS

In the following, we show that non-invertible transformations change the resulting distribution of the transformed representation in the MAP framework and introduce the family of projections on $L_p$ hyperspheres.

### 4.1.1 INVERTIBLE TRANSFORMATIONS

In the above, we have modeled the conditional distribution $f_c(.)$ for a class $c$ through the intermediate variable $z \in \mathbb{R}^L$, the neural network output. Let us now consider that an additional transformation $h : \mathbb{R}^L \to \mathbb{R}^L$ is applied to $z$. If $h(.)$ is a one-to-one invertible transformation, the conditional probability density function $q_c$ of the resulting variable $\boldsymbol{\zeta} = h(z)$ can be expressed as in Equation 8 Murphy (2022).

$$q_c(\boldsymbol{\zeta}) = \frac{f_c(z)}{|J_h(\boldsymbol{\zeta})|} \tag{8}$$

With $J_h(\boldsymbol{\zeta})$ the Jacobian of $h$ and $|J_h(\boldsymbol{\zeta})|$ its determinant evaluated at $\boldsymbol{\zeta}$. Starting from Equation 2, it follows that trying to express the posterior $p(c|\boldsymbol{\zeta})$ with regard to $\boldsymbol{\zeta}$ leads to:

$$p(c|\boldsymbol{\zeta}) = \frac{q_c(\boldsymbol{\zeta})\pi_c}{\sum_{\ell=1}^{L} q_\ell(\boldsymbol{\zeta})\pi_\ell} = \frac{\frac{f_c(z)}{|J_h(\boldsymbol{\zeta})|}\pi_c}{\sum_{\ell=1}^{L} \frac{f_\ell(z)}{|J_h(\boldsymbol{\zeta})|}\pi_\ell} = p(c|\boldsymbol{z}) \tag{9}$$

Thus, combining invertible transformations with the MAP framework gives strictly identical a posteriori probability distributions. This observation also holds for Cross-Entropy given the equivalence showed in Section 3.3.

### 4.1.2 PROJECTIONS ON $L_p$ HYPERSPHERES

This family of transformations reduces the vector's dimensionality, resulting in a non-invertible transformation. We define such transformations as $T_{l_p} : \mathbb{R}^L \to \mathbb{R}^L$ on $\boldsymbol{z} = (z_1, \cdots, z_L) \in \mathbb{R}^L$ such that:

$$T_{l_p}(\boldsymbol{z}) = \frac{z}{||\boldsymbol{z}||_p}, \tag{10}$$

with $\boldsymbol{z}$ the output representation of the neural network, $||\boldsymbol{z}||_p = (\sum_{i=1}^{L} |z_i|^p)^{1/p}$ and $|z_i|$ the absolute value of $z_i$.

## 4.2 PROJECTED GAUSSIAN DISTRIBUTION

Given the previous analysis, we argue that using SCE-$\tau$ on projected representation is not theoretically justified. Indeed, the result of a radial projection (equivalently, normalization) of a Gaussian distribution on the $L_p$ unit hypersphere is most likely not Gaussian. Additionally, such transformation being non invertible, the obtained MAP objective should be adapted accordingly. We propose an analytical expression for the projection of a Gaussian distribution on any $L_p$ hypersphere and give the proof of this result in Appendix B.

**Proposition 4.1.** *Let $p, d \in \mathbb{N}^{+\star}$. For $\boldsymbol{z} \in \mathbb{R}^d$ following a d-variate Gaussian of mean $\boldsymbol{\mu} \in \mathcal{S}_p^d$ and covariance matrix $\Sigma = \sigma^2 I$, the distribution of $\boldsymbol{u}$, the projection of $\boldsymbol{z}$ on $\mathcal{S}_p^d$ such that $\boldsymbol{u} = \frac{\boldsymbol{z}}{||\boldsymbol{z}||_p}$ is defined by:*

$$g_\kappa^{PGD}(\boldsymbol{u}, \boldsymbol{\mu_c}) = a_\kappa e^{-\frac{1}{2}\kappa^2} \sum_{n=0}^{\infty} \frac{(\kappa \frac{\boldsymbol{u}^T \cdot \boldsymbol{\mu}}{||\boldsymbol{u}||_2 \cdot ||\boldsymbol{\mu}||_2})^n \, \Gamma\left(\frac{d}{2} + \frac{n}{2}\right)}{n! \, \Gamma\left(\frac{d}{2}\right)} \tag{11}$$

*with $\kappa^2 = \frac{||\boldsymbol{\mu}||_2}{\sigma^2}$, $a_\kappa = \frac{\Gamma\left(\frac{d}{2}\right)\left(\boldsymbol{u}^T \boldsymbol{u}\right)^{-\frac{d}{2}}}{2\pi^{\frac{d}{2}} w}$ a normalization factor and $w = ||u||_{2(p-1)}^{(p-1)}$*

## 4.3 PGD-LOSS EXPRESSION

We define $\mathcal{L}_{PGD}^p$ on the standard simplex by combining PGD from equation 11 and the MAP log-loss from equation 5:

$$\mathcal{L}_{PGD}^p(\mathcal{B}, \theta) = -\frac{1}{|\mathcal{B}|} \sum_{c=1}^{L} \sum_{i \in I_c} \log \frac{g_\kappa^{PGD}(T_{l_p}(\Phi_\theta(\mathbf{x}_i)), \boldsymbol{e_c})}{\sum_{\ell=1}^{L} g_\kappa^{PGD}(T_{l_p}(\Phi_\theta(\mathbf{x}_i)), \boldsymbol{e_\ell})} \tag{12}$$

## 4.4 SCE-$\tau$ ON $L_p$ HYPERSPHERE

SCE-$\tau$ can be used with representations projected onto the $L_p$ hypersphere, even though the Gaussian assumption is not fulfilled. Various works have shown that SCE-$\tau$ can empirically achieve competitive performances on the $L_2$ hypersphere De Brebisson & Vincent (2015); Wang et al. (2017). In our setting, a potential justification of such results is the validity of a Gaussian projection approximation for small variance values. Indeed, the projection of a multivariate Gaussian along one of its components is a Gaussian. We refer to this projection as an axial projection. While such a result does not hold for radial projections (or normalizations), we can show that the radial and axial projections tend to result in the same projections when $v$ tends to 0. Let us consider $\boldsymbol{z} = [z_1, \cdots, z_L] \in \mathbb{R}^L$, a vector sampled from a Gaussian centered around $\boldsymbol{e_1} = [1, 0 \cdots, 0] \in \mathbb{R}^L$. It follows:

$$\boldsymbol{z}_p = \left[\frac{z_1}{||\boldsymbol{z}||_p}, \frac{z_2}{||\boldsymbol{z}||_p}, \cdots, \frac{z_L}{||\boldsymbol{z}||_p}\right] \text{ and } \boldsymbol{z}_a = [1, z_2, \cdots, z_L] \tag{13}$$

with $\boldsymbol{z}_p$ and $z_a$ being the radial $L_p$ and axial projections respectively. It follows that

$$\begin{cases} \boldsymbol{z} & \to \boldsymbol{e_1} \\ v & \to 0 \end{cases} \quad \begin{cases} \boldsymbol{z}_p & \to \boldsymbol{e_1} \\ v & \to 0 \end{cases} \quad \begin{cases} ||\boldsymbol{z}_p - \boldsymbol{z}_a||_2 & \to 0 \\ v & \to 0 \end{cases} \tag{14}$$

Hence, the smaller the variance, the more likely axial and radial projections will lead to the same resulting Gaussian distribution. A geometric interpretation is that for small variance values, the hypersphere surface around the mean can be approximated by a plane perpendicular to the mean direction. Of course, such approximation differs for different values of $p$, in the case of $p = \infty$, the surface is a plane perpendicular to the mean. In the case of $p = 2$, the surface might be considered planar locally. Hence, we expect the optimal value of $v$ when training with SCE-$\tau$ to be proportional to the value of $p$.

## 4.5 PROJECTION EQUIVALENCE

In Section 4.2, we have given an expression of the PGD on any $L_p$ hypersphere. Remarkably, changing the value of $p$ only impacts the normalization term $a_\kappa$. Indeed, for the other term depending

on $\boldsymbol{u}$, denoting $\boldsymbol{u_p} = \frac{\boldsymbol{u}}{||\boldsymbol{u}||_p}$, we have for any $p \in \mathbb{N}^{+\star}$:

$$\frac{\boldsymbol{u_p}^T \cdot \boldsymbol{\mu}}{||\boldsymbol{u_p}||_2 \cdot ||\boldsymbol{\mu}||_2} = \frac{\frac{\boldsymbol{u}}{||\boldsymbol{u}||_p}^T \cdot \boldsymbol{\mu}}{||\frac{\boldsymbol{u}}{||\boldsymbol{u}||_p}||_2 \cdot ||\boldsymbol{\mu}||_2} = \frac{\boldsymbol{u}^T \cdot \boldsymbol{\mu}}{||\boldsymbol{u}||_2 \cdot ||\boldsymbol{\mu}||_2} \tag{15}$$

Therefore, when plugging the expression of PGD from equation 11 into the MAP log-loss expression from equation 5, the normalization factors simplify, and the resulting loss is unchanged, no matter the value of $p$ used when projecting. It follows that, in the MAP framework, every projection is equivalent to a projection on the unit hypersphere with the correct probabilistic modeling. As discussed in section 4.1.1, this result was predictable as every projection on the $L_p$ unit sphere can be deduced from another through an invertible transformation. This is true in our MAP framework since the normalization term $a_\kappa$ is de facto ignored.

## 5 EXPERIMENTS

In the following section, we conduct experiments on standard computer vision datasets for image classification. We compare the performances of SCE, SCE-$\tau$, and the loss function derived from MAP with the PGD model and confirm our intuitions based on our insights from the MAP modeling.

### 5.1 EXPERIMENTAL SETUP

**Datasets.** To compare the presented losses, we use 3 benchmark datasets. CIFAR10 Krizhevsky (2009) is composed of 50,000 train images and 10,000 test images for 10 classes. All images are of size 32×32. CIFAR100 Krizhevsky (2009) is similarly composed of 50,000 32×32 train images and 10,000 test images but has 100 classes. Imagenet100 is a subset of the ILSVRC-2012 Deng et al. (2009) classification dataset. Different from Tiny-ImagNet, ImageNet100 is composed of the 100 first classes of ILSVRC-2012. This corresponds to a total of 130,000 224x224 train images and 5,000 224x224 test images.

**Losses and projections.** In these experiments, we compare the performances of the following losses: SCE, SCE-$\tau$ and PGD-loss. Additionally, we compare projections on various $L_p$ hyperspheres with $p \in \{0.5, 1, 2, 3, \infty\}$.

**Implementation details** For each loss, we train a ResNet18 He et al. (2016) from scratch for 300 epochs with an Adam Kingma & Ba (2014) optimizer, learning rate $1e^{-4}$, and a batch size value of 256. We also use data augmentations. Namely, random horizontal flip, random crop and color jitter. The main results showed in Table 1 have been obtained with the best variance values after conducting a hyper-parameter search. More details can be found in Appendix D.

### 5.2 RESULTS

**Accuracy.** Table 1 shows the obtained accuracy at the end of training for SCE, SCE-$\tau$ and PGD losses on considered datasets. The value of $p$ indicates the hypersphere on which representations are projected. For baseline, performances of SCE and SCE-$\tau$ without projection are also reported. Following previous studies, it can be observed that projecting representations on the $L_2$ hypersphere leads to a significant increase in performance, given that the optimal variance (or equivalently temperature) is used. Furthermore, we observe that similar performances can be obtained on all datasets for any projection strategies with SCE-$\tau$. Eventually, the obtained results with PGD are on par with the best SCE-$\tau$ results. In the case of PGD, we indicate no values of $p$ since the loss is independent of the projection strategy.

**Impact of $v$.** We study the impact of the variance parameter for SCE-$\tau$ and PGD losses on CIFAR10 and CIFAR100. Figure 2 shows the accuracy at the end of training with SCE-$\tau$ on CIFAR10 for various values of $p$ and $v$. For each value of $p$, an optimal value of $v$ can be found to obtain the best performances. Notably, a strong performance degradation occurs for large variances rather than for smaller variances. However, a lower variance value might hinder training stability. Additionally, according to the intuition given in Section 4.4 and similar to the results presented in Table 1, the

| Loss | $p$ | CIFAR10 | | CIFAR100 | | ImageNet100 | |
|------|-----|---------|---|----------|---|-------------|---|
| | | Acc. | $v$ | Acc. | $v$ | Acc. | $v$ |
| SCE | no proj. | 90.44±0.44 | N/A | 65.44±0.64 | N/A | 63.38 | N/A |
| SCE-$\tau$ | no proj. | 90.93±0.31 | 2.3 | 66.20±0.69 | 2.7 | 64.16 | 2.7 |
| SCE-$\tau$ | $p = 0.5$ | 92.15±0.19 | 0.006 | 68.56±0.33 | 5e-05 | 66.52 | 5e-05 |
| SCE-$\tau$ | $p = 1$ | 92.48±0.13 | 0.15 | 68.62±0.38 | 0.007 | 65.84 | 0.007 |
| SCE-$\tau$ | $p = 1.5$ | 92.32±0.30 | 0.30 | 68.19±0.45 | 0.035 | 67.32 | 0.025 |
| SCE-$\tau$ | $p = 2$ | 92.14±0.21 | 0.45 | 68.67±0.48 | 0.050 | 67.34 | 0.050 |
| SCE-$\tau$ | $p = 3$ | 92.22±0.45 | 0.50 | 68.90±0.30 | 0.09 | 66.98 | 0.09 |
| SCE-$\tau$ | $p = \infty$ | 91.91±0.27 | 0.40 | 68.69±0.37 | 0.22 | 67.16 | 0.22 |
| PGD | any | 92.36±0.26 | 0.35 | 68.84±0.18 | 0.12 | 66.30 | 0.21 |

Table 1: Accuracy (%) of different losses and projections strategies on CIFAR10, CIFAR100, and ImageNet. SCE corresponds to Softmax Cross-Entropy and SCE-$\tau$ corresponds to SCE with temperature and PGD to the PGD loss defined in 12. The values of $p$ and $v$ used for training are similarly reported. For CIFAR10 and CIFAR100, the average and standard deviation over 5 runs are reported, while only 1 run was realised for ImageNet100.

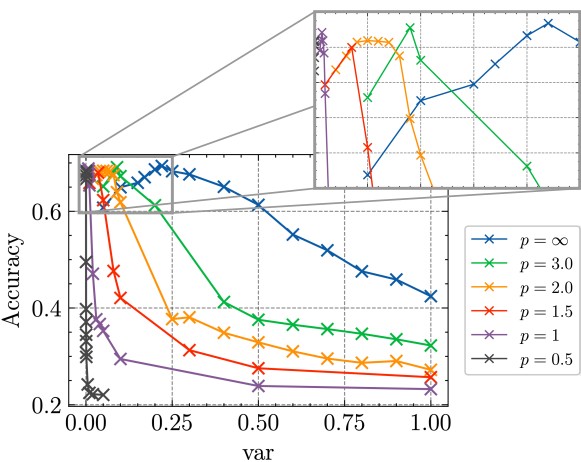

Figure 2: Accuracy at the end of training a ResNet18 on CIFAR100 with a MAP objective (or equivalently SCE-$\tau$) for different $(p, v)$ values. The top left part is zoomed in for better readability.

larger the value of $p$, the greater the resulting optimal variance is. Plus, SCE-$\tau$ performances gain in stability with regard to $v$ for larger values of $p$. We discuss this phenomenon in more detail in Section 5.3. Moreover, Figure 3 shows the final accuracy when training with SCE-$\tau$ on CIFAR10, and comparable observations as on CIFAR100 can be made.

Since the proposed PGD loss is invariant with $p$, Figure 4 shows only the impact of $v$ on the final accuracy when training with PGD. Notably, PGD exhibits similar performances to SCE-$\tau$ with $p = \infty$, not only in terms of maximum performance but also in terms of stability with regard to $v$. We discuss such similarity in Section 5.3.

## 5.3 DISCUSSIONS

From the results presented above, we make the following observations. 1) Similar results can be obtained for SCE-$\tau$ and PGD losses on any $L_p$ hypersphere. We believe this to be a direct consequence of Gaussian projections tending to be Gaussian for smaller values of $v$. In this situation, SCE-$\tau$ becomes a valid theoretical objective to minimize since we showed its equivalence to the MAP log-loss objective. In that sense, given the appropriate variance is used, SCE-$\tau$ and PGD losses should give similar final solutions, hence the obtained accuracies. 2) The sensitivity to $v$ in term of accuracy is larger for smaller values of $p$. We believe this to be a consequence of the resulting flatness of the $L_p$ hypersphere around the mean vector. When $p < 1$, the obtained shape is an astroid whose shape is particularly sharp around the standard basis vectors. When $p = \infty$, the resulting shape is a

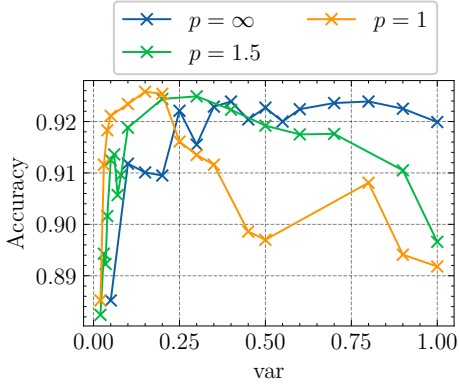
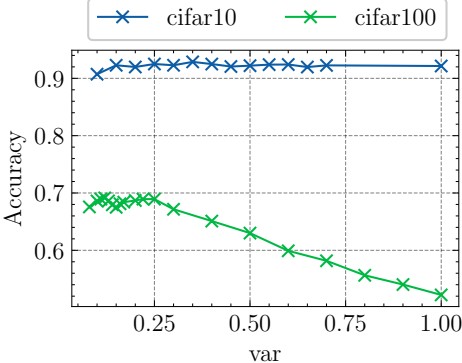

Figure 3: Accuracy at the end of training on CIFAR10 with a SCE-$\tau$ objective for different values of $p$ and variance.

Figure 4: Accuracy after training a ResNet18 on CIFAR100 and CIFAR10 with the PGD for different values of $v$.

hypercube where each face is centered around a vector of the standard basis. In that case, the $L_p$ is exactly planar locally, even when moving further from the mean. In other words, large values of $p$ make an easier approximation of the projection as Gaussian, and the larger the value of $p$, the more this approximation holds even for greater variances. 3) The PGD loss is more stable than the SCE-$\tau$ loss with regard to $v$ if $p \neq \infty$ and displays similar stability when $p = \infty$. Since PGD is the resulting distribution from a radial projection, the Gaussian approximation is not necessary, and the model remains valid even for larger values of $v$. However, a performance drop is still observed when $v$ is getting too large, notably on CIFAR100. Even with a more accurate estimation of the projected distribution, when $v$ is too large, the model might not be discriminative enough for the classification task due to excessive overlap in the modeled Gaussian. Eventually, as discussed above, when $p = \infty$ the approximation of the projected as a Gaussian is the most valid when compared to lower values of $p$. In that sense, PGD and SCE-$\tau$ present similar behaviour since both are sound modeling. On top of the previously listed advantages, the infinite norm is extremely simple and stable to compute and should be considered as an alternative to the $L_2$ norm when training DNNs.

## 6 CONCLUSION

This paper provides a unified perspective on the connection between output normalization and loss functions in classification problems. By extending the Maximum-a-Posteriori (MAP) approach to encompass both the loss function and output normalization, we have established theoretical connections between the Softmax Cross-entropy (SCE) and its variants, including SCE-$\tau$ and SCE with prototype and bias. Our results demonstrate that SCE-$\tau$ can be interpreted as a MAP with a class-conditional isotropic Gaussian hypothesis on the standard simplex and that the temperature can be expressed as the ratio between, the scaling factor and the variance of Gaussian distributions. However, we indicated that such an objective is not theoretically adapted when projecting on the $L_p$ hypersphere. Therefore, we have introduced the Projected Gaussian Distribution (PGD) to model Gaussian distributions projected on any $L_p$ hypersphere. We showed that in our framework, projections on $L_p$ hyperspheres are equivalent for all values of $p$. Moreover, we showed that even though SCE-$\tau$ cannot be justified by our theory in general, it is a valid approximation for small variance values. Finally, we give evidence that PGD and SCE-$\tau$ on the hypercube present several advantages over other values of $p$, such as greater stability with respect to $v$ and computational simplicity in the case of the hypercube.

Eventually the modeling is based on the assumption that the network outputs can be approximated by a Gaussian distribution; which can be a limitation in some specific cases. Presented performances and comparisons are established with a specific DNN and problem setting (image classification); of course, different figures can be obtained with other settings. As a future study, we plan to investigate training with a maximum likelihood approach equipped with PGD, or considering mixtures other than Gaussian. Another research topic would be exploring prototype learning on $L_p$ hyperspheres.

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

## A  PROOF OF PROPOSITION 3.1

Starting with Equation 5, the conditional probability distribution of $Z$ given $Y = c$ follows a Gaussian distribution centered around $\boldsymbol{r_c} \in \mathbb{R}^L$, with covariance matrix $\Sigma_c$:

$$f_c(\mathbf{z}) = (2\pi)^{-L/2}|\Sigma_c|^{-1}e^{-\frac{1}{2}(\mathbf{z}-\boldsymbol{r_c})^T\Sigma_c^{-1}(\mathbf{z}-\boldsymbol{r_c})}, \tag{16}$$

with $T$ being the superscript for the transpose operator and $|\Sigma_c|$ the determinant of $\Sigma_c$. The conditional Gaussian are isotropic if $\Sigma_c = v_c \cdot I$ with $I$ being the identity matrix of size $L$ and $v_c$ the variance for class $c$. In such situation, $f_c(.)$ becomes

$$f_c(\mathbf{z}) = (2\pi v_c)^{-L/2}e^{-\frac{1}{2v_c}||\mathbf{z}-\boldsymbol{r_c}||_2^2} \tag{17}$$

Combining Equations equation 5 and equation 17 leads to the general form below.

$$
\begin{aligned}
\mathcal{L}_{Gauss}(\mathcal{B},\theta) &= -\frac{1}{|\mathcal{B}|}\sum_{c=1}^{L}\sum_{i\in I_c}\log\frac{\pi_c \cdot (2\pi v_c)^{-L/2}e^{-\frac{1}{2v_c}||\Phi_\theta(\mathbf{x}_i)-\boldsymbol{r_c}||_2^2}}{\displaystyle\sum_{\ell=1}^{L}\pi_l \cdot (2\pi v_l)^{-L/2}e^{-\frac{1}{2v_l}||\Phi_\theta(\mathbf{x}_i)-\boldsymbol{r_l}||_2^2}} \\
&= -\frac{1}{|\mathcal{B}|}\sum_{c=1}^{L}\sum_{i\in I_c}\log\frac{\pi_c \cdot v_c^{-L/2}e^{\frac{1}{v_c}\Phi_\theta(\mathbf{x}_i)^T\cdot\boldsymbol{r_c}-\frac{1}{2v_c}||\Phi_\theta(\mathbf{x}_i)||_2^2-\frac{1}{2v_c}||\boldsymbol{r_c}||_2^2}}{\displaystyle\sum_{\ell=1}^{L}\pi_l \cdot v_l^{-L/2}e^{\frac{1}{v_l}\Phi_\theta(\mathbf{x}_i)^T\cdot\boldsymbol{r_l}-\frac{1}{2v_l}||\Phi_\theta(\mathbf{x}_i)||_2^2-\frac{1}{2v_l}||\boldsymbol{r_l}||_2^2}}
\end{aligned} \tag{18}
$$

Now, with equal variances, previous Equation equation 18 simplifies to:

$$\mathcal{L}_{Gauss}(\mathcal{B},\theta) = -\frac{1}{|\mathcal{B}|}\sum_{c=1}^{L}\sum_{i\in I_c}\log\frac{\pi_c \cdot e^{\frac{1}{v}\Phi_\theta(\mathbf{x}_i)^T\cdot\boldsymbol{r_c}}}{\displaystyle\sum_{\ell=1}^{L}\pi_l \cdot e^{\frac{1}{v}\Phi_\theta(\mathbf{x}_i)^T\cdot\boldsymbol{r_l}}} \tag{19}$$

The means are assigned to the re-scaled standard basis vectors such that $\boldsymbol{r_c} = r \cdot \boldsymbol{e_c}$ with $e_c = [0,0,\dots,1,0,\dots0]$ a vector where every component is 0 except the c-th component and $c \in [\![1,L]\!]$. Therefore, the previous equation can be rewritten like in Equation 20 and this ends the proof:

$$
\begin{aligned}
\mathcal{L}_{Gauss}(\mathcal{B},\theta) &= -\frac{1}{|\mathcal{B}|}\sum_{c=1}^{L}\sum_{i\in I_c}\log\frac{\pi_c \cdot e^{\frac{r}{v}\Phi_\theta(\mathbf{x}_i)^T\cdot\boldsymbol{e_c}}}{\displaystyle\sum_{\ell=1}^{L}\pi_l \cdot e^{\frac{r}{v}\Phi_\theta(\mathbf{x}_i)^T\cdot\boldsymbol{e_l}}} \\
&= -\frac{1}{|\mathcal{B}|}\sum_{c=1}^{L}\sum_{i\in I_c}\log\frac{\pi_c \cdot e^{\frac{r}{v}\Phi_\theta(\mathbf{x}_i)_c}}{\displaystyle\sum_{\ell=1}^{L}\pi_l \cdot e^{\frac{r}{v}\Phi_\theta(\mathbf{x}_i)_l}}
\end{aligned} \tag{20}
$$

## B  PROOF OF PROPOSITION 4.1

Let $\boldsymbol{z}$ be a random vector of $\mathbb{R}^d$ with a Gaussian distribution of mean $\boldsymbol{\mu}$ and covariance matrix $\Sigma$:

$$f_Z(\boldsymbol{z}) = \frac{1}{(2\pi)^{\frac{d}{2}}|\Sigma|^{\frac{1}{2}}}\exp\left(-\frac{1}{2}(\boldsymbol{z}-\boldsymbol{\mu})^T\Sigma^{-1}(\boldsymbol{z}-\boldsymbol{\mu})\right) \tag{21}$$

and define

$$\boldsymbol{u} = \frac{\boldsymbol{z}}{||\boldsymbol{z}||_p} = \frac{\boldsymbol{z}}{||\boldsymbol{z}||_p} = \frac{\boldsymbol{z}}{r} \tag{22}$$

the projected vector onto the unit sphere $S_p^d = \{\boldsymbol{x}\in\mathbb{R}^d : ||\boldsymbol{x}||_p = 1\}$. The marginal of $z$ on $S_2^d$ is called *projected-normal* in Jupp & Mardia (2009).

We present several expressions for the density function $f_U(\boldsymbol{u})$ of the normalized vector $\boldsymbol{u}$. Building on previous work by Pukkila & Radhakrishna Rao (1988) and extending the result to general cases

where $p \neq 2$, we provide a recursively computable integral representation, proving a result which has been stated inSaw (1978) without direct proof. Furthermore, we derive a closed-form expression in terms of a special function. To begin with, we establish a change-of-variable formula for $z \to (r, u)$, where $u$ is constrained to live in $\mathcal{S}_p^d$. Let $r = ||\boldsymbol{z}||_p$. We begin with a result on the change of variable $z \to (r, u)$, where $u$ is constrained to live in $\mathcal{S}_p^d$.

**Proposition B.1.** *If $z$ has a probability density $f_Z(z)$, with $z \in \mathbb{R}^d$, then the transformation $z \to (r, u)$, where $u$ is constrained to live in $\mathcal{S}_p^d$ leads to the density $f_{R,U}(r, u)$:*

$$f_{R,U}(r, u) = \frac{r^{d-1}}{||u||_{2(p-1)}^{p-1}} f_Z(r.u) \tag{23}$$

*with respect to $d_\sigma$, the element of area of the surface $\mathcal{S}_p^d$.*

*Proof.* Let $\xi = \Phi(z_1, \cdots, z_d)$ define a surface element in $\mathbb{R}^d$. A general result in Courant (2011) pages 301-302, states that for any function, we have

$$\int \cdots \int f(z_1, \cdots, z_d) dz_1 \cdots dz_d = \int \cdots \int \frac{f(z_1, \cdots, z_d)}{\sqrt{\Phi_{z_1}^2 + \cdots + \Phi_{z_d}^2}} d_{\sigma_\xi} d\xi$$

where $\Phi_{z_i} = \frac{\delta \Phi}{\delta z_i}$ and $d_{\sigma_\xi} = \frac{\sqrt{\Phi_{z_i}^2 + \cdots + \Phi_{z_d}^2}}{\Phi_{z_d}} dz_1 \cdots dz_{d-1}$ with $\Phi(z_1, \cdots, z_d) = \Sigma_{i=1}^d |z_i|^p = ||z||_p^p$, we have

$$\sqrt{\Phi_{z_1}^2 + \cdots + \Phi_{z_d}^2} = \sqrt{\Sigma_{i=1}^d \left(p|z_i|^{p-1} sign(z_i)\right)^2} \tag{24}$$

$$= p\sqrt{||z||_{2(p-1)}^{2(p-1)}} = p||z||_{2(p-1)}^{p-1} \tag{25}$$

with $\xi = r^p$, we have $d\xi = d(r^p) = pr^{p-1}dr$.

Now, if we let $z = ru$, it becomes clear that $d_{\sigma_r} = r^{d-1} d_\sigma$, where $d_\sigma$ is the element of area of $\mathcal{S}_p^d$ and $d_{\sigma_r}$ is the element of area of the surface $||.||_p = r$. On the other hand, we have $||z||_p^{p-1} = r^{p-1}||u||_p^{p-1}$. Combining these elements, we obtain:

$$f_Z(z_1, \cdots, z_d) d_z = f_{R,U}(r, u) dr d_\sigma = \frac{r^{d-1}}{||u||_{2(p-1)}^{p-1}} f_Z(r.u) dr d\sigma \tag{26}$$

which gives the result.

**Remark B.2.** *Observe that with $p = 2$, $||u||_{2(p-1)}^{(p-1)} = ||u||_2^1 = 1$ and $f_{R,U}(r, u) = r^{d-1} f_Z(ru)$.*

$\square$

**Proposition B.3.** *The projection of a normal distribution on $S_p^d$ is:*

$$f_U(\boldsymbol{u}) = \frac{(\boldsymbol{u}^T \Sigma^{-1} \boldsymbol{u})^{-\frac{d}{2}}}{(2\pi)^{\frac{d}{2}} |\Sigma|^{\frac{1}{2}} w} \exp\left(-\frac{1}{2}\lambda^2\right) \int_0^\infty r'^{d-1} \exp\left(-\frac{1}{2}r'^2 + \lambda r' \, \bar{\boldsymbol{u}}^T \Sigma^{-1} \bar{\boldsymbol{\mu}}\right) \mathrm{d}r' \tag{27}$$

*with $\lambda = (\boldsymbol{\mu}^T \Sigma^{-1} \boldsymbol{\mu})^{\frac{1}{2}}$, $\bar{\boldsymbol{u}} = \frac{\boldsymbol{u}}{(\boldsymbol{u}^T \Sigma^{-1} \boldsymbol{u})^{\frac{1}{2}}}$, $w = ||u||_{2(p-1)}^{(p-1)}$ and $\bar{\boldsymbol{\mu}} = \frac{\boldsymbol{\mu}}{(\boldsymbol{\mu}^T \Sigma^{-1} \boldsymbol{\mu})^{\frac{1}{2}}}$*

*Proof.* By a direct application, we get the density for a normal distribution:

$$f_{R,U}(r, \boldsymbol{u}) = \frac{r^{d-1}}{(2\pi)^{\frac{d}{2}} |\Sigma|^{\frac{1}{2}} w} \exp\left(-\frac{1}{2}(r\boldsymbol{u} - \boldsymbol{\mu})^T \Sigma^{-1} (r\boldsymbol{u} - \boldsymbol{\mu})\right)$$

$$= \frac{r^{d-1}}{(2\pi)^{\frac{d}{2}} |\Sigma| w} \exp\left(-\frac{1}{2}\boldsymbol{\mu}^T \Sigma^{-1} \boldsymbol{\mu}\right) \exp\left(-\frac{1}{2}r^2 \boldsymbol{u}^T \Sigma^{-1} \boldsymbol{u} + r\boldsymbol{u}^T \Sigma^{-1} \boldsymbol{\mu}\right). \tag{28}$$

with $w = ||u||_{2(p-1)}^{(p-1)}$. The density for $f_U(\boldsymbol{u})$ is obtained by marginalizing $f_{R,U}(r, \boldsymbol{u})$ over $r$: $f_U(\boldsymbol{u}) = \int_0^\infty f_{R,U}(r, \boldsymbol{u}) \mathrm{d}r$. Let $r' = r(\boldsymbol{u}^T \Sigma^{-1} \boldsymbol{u})^{\frac{1}{2}}$; then

$$f_U(\boldsymbol{u}) = \frac{(\boldsymbol{u}^T \Sigma^{-1} \boldsymbol{u})^{-\frac{d}{2}}}{(2\pi)^{\frac{d}{2}} |\Sigma|^{\frac{1}{2}} w} \exp\left( -\frac{1}{2} \boldsymbol{\mu}^T \Sigma^{-1} \boldsymbol{\mu} \right) \int_0^\infty r'^{d-1} \exp\left( -\frac{1}{2} r'^2 + r' \frac{\boldsymbol{u}^T \Sigma^{-1} \boldsymbol{\mu}}{\boldsymbol{u}^T \Sigma^{-1} \boldsymbol{u}} \right) \mathrm{d}r' \quad (29)$$

Denoting $\lambda = (\boldsymbol{\mu}^T \Sigma^{-1} \boldsymbol{\mu})^{\frac{1}{2}}$, $\bar{\boldsymbol{u}} = \frac{\boldsymbol{u}}{(\boldsymbol{u}^T \Sigma^{-1} \boldsymbol{u})^{\frac{1}{2}}}$ and $\bar{\boldsymbol{\mu}} = \frac{\boldsymbol{\mu}}{(\boldsymbol{\mu}^T \Sigma^{-1} \boldsymbol{\mu})^{\frac{1}{2}}}$, which finally gives equation 29.
$\square$

**Remark B.4.** With $p = 2, \boldsymbol{\mu} = 0$ and $\Sigma = \sigma^2 1$, which means that $x$ is distributed as a centered isotropic Gaussian, equation 27 reduces to

$$f_U(u) = \frac{1}{(2\pi)^{\frac{d}{2}}} \int_0^\infty r'^{d-1} \exp\left( -\frac{1}{2} r'^2 \right) \mathrm{d}r' = \frac{\Gamma\left(\frac{d}{2}\right)}{2\pi^{\frac{d}{2}}} = \frac{1}{\omega_{d-1}} \quad (30)$$

where we used $u^T u = 1$ and the known property

$$\int_0^\infty r^{d-1} \exp\left( -\frac{1}{2} r^2 \right) \mathrm{d}r = 2^{\frac{d}{2}-1} \Gamma\left(\frac{d}{2}\right). \quad (31)$$

Equation equation 30 shows that $f_U(u)$ is the uniform distribution on the unit-sphere, where $\omega_{d-1}$ is the surface of the unit-sphere.

Starting with equation 29, we can now state the first result, which is due to Pukkila & Radhakrishna Rao (1988).

**Proposition B.5.** With $\lambda = (\boldsymbol{\mu}^T \Sigma^{-1} \boldsymbol{\mu})^{\frac{1}{2}}$ and $\alpha = \frac{u^T \Sigma^{-1} \boldsymbol{\mu}}{u^T \Sigma^{-1} u}$, the probability density of the normalized Gaussian vector is

$$f_U(u) = \frac{(u^T \Sigma^{-1} u)^{-\frac{d}{2}}}{(2\pi)^{\frac{d}{2}-1} |\Sigma|^{\frac{1}{2}} w} \exp\left( -\frac{1}{2} \left( \lambda^2 - \alpha^2 \right) \right) I_d(\alpha) \quad (32)$$

with

$$I_d(\alpha) = \frac{1}{\sqrt{2\pi}} \int_0^\infty r^{d-1} \exp\left( -\frac{1}{2} (r - \alpha)^2 \right) \mathrm{d}r \quad (33)$$

and can be computed as

$$I_d(\alpha) = \alpha I_{d-1}(\alpha) + (d-2) I_{d-2}(\alpha),$$

with $I_1 = \Phi(\alpha)$ and $I_2 = \phi(\alpha) + \alpha \Phi(\alpha)$, where $\phi(.)$ and $\Phi(.)$ are respectively the standard normal probability density function and cumulative distribution function.

*Proof.* Completing the square in the argument of the exponential under the integral in equation 29 gives equation 32, with the definition of $I_d$ in equation 33. Integration by part of $I_d$ yields the recurrence equation. Finally, the initial values follow by direct calculation. $\square$

The main drawback of Equation equation 32 is that it relies on an integral form, although this integral can be easily evaluated through a recurrence. In contrast, Equation equation 27 allows us to express the density as a series. We present this result in the general case and recover the result stated in Saw (1978) without proof.

**Proposition B.6.** With $\lambda = (\boldsymbol{\mu}^T \Sigma^{-1} \boldsymbol{\mu})^{\frac{1}{2}}$, $\bar{u} = \frac{u}{(u^T \Sigma^{-1} u)^{\frac{1}{2}}}$, $\bar{\boldsymbol{\mu}} = \frac{\boldsymbol{\mu}}{(\boldsymbol{\mu}^T \Sigma^{-1} \boldsymbol{\mu})^{\frac{1}{2}}}$, $w = ||u||_{2(p-1)}^{(p-1)}$ the probability density of the normalized Gaussian vector is

$$f_U(u) = \frac{\Gamma\left(\frac{d}{2}\right)}{2\pi^{\frac{d}{2}}} \frac{(u^T \Sigma^{-1} u)^{-\frac{d}{2}}}{|\Sigma|^{\frac{1}{2}} w} e^{-\frac{1}{2}\lambda^2} \sum_{k=0}^\infty \left( \lambda \bar{u}^T \Sigma^{-1} \bar{\boldsymbol{\mu}} \right)^k \frac{\Gamma\left(\frac{d+k}{2}\right)}{k! \, \Gamma\left(\frac{d}{2}\right)} \quad (34)$$

*Proof.* In the integral in equation 27, we can expand the exponential $\exp\left(\lambda r\,\bar{u}^T\Sigma^{-1}\bar{\boldsymbol{\mu}}\right)$ in Taylor series, so that

$$
\int_0^\infty r^{d-1}\exp\left(-\frac{1}{2}r^2 + \lambda r\,\bar{u}^T\Sigma^{-1}\bar{\boldsymbol{\mu}}\right)\mathrm{d}r
$$

$$
= \int_0^\infty r^{d-1}\exp\left(-\frac{1}{2}r^2\right)\sum_{k=0}^\infty \frac{1}{k!}\left(\lambda r\,\bar{u}^T\Sigma^{-1}\bar{\boldsymbol{\mu}}\right)^k\mathrm{d}r
$$

$$
= \sum_{k=0}^\infty \frac{1}{k!}\left(\lambda\bar{u}^T\Sigma^{-1}\bar{\boldsymbol{\mu}}\right)^k \int_0^\infty r^{d-1+k}\exp\left(-\frac{1}{2}r^2\right) \tag{35}
$$

$$
= 2^{\frac{d}{2}-1}\sum_{k=0}^\infty \frac{1}{k!}\left(\lambda\bar{u}^T\Sigma^{-1}\bar{\boldsymbol{\mu}}\right)^k \Gamma\left(\frac{d+k}{2}\right)
$$

where the last line follows from the identity equation 31. Plugging this in equation 27 and simplifying yield equation 34. $\qquad\square$

For $p = 2$, we can observe that the first term in equation 34 is the inverse of the unit-sphere's surface $\omega_{d-1}$. Still for $= 2$, in the isotropic case where $\Sigma = \sigma^2\mathbb{1}$, equation 34 reduces to

$$
f_U(u) = \frac{\Gamma\left(\frac{d}{2}\right)}{2\pi^{\frac{d}{2}}}\,e^{-\frac{1}{2}\lambda^2}\sum_{k=0}^\infty \left(\lambda u^T\bar{\boldsymbol{\mu}}\right)^k \frac{\Gamma\left(\frac{d+k}{2}\right)}{k!\,\Gamma\left(\frac{d}{2}\right)} \tag{36}
$$

where we used the fact that $u^T u = 1$ and where $\bar{\boldsymbol{\mu}}$ is now $\bar{\boldsymbol{\mu}} = \frac{\boldsymbol{\mu}}{(\boldsymbol{\mu}^T\boldsymbol{\mu})^{\frac{1}{2}}}$. This is the formula given in Saw (1978), up to minor notations differences. Finally, for $\boldsymbol{\mu} = 0$, equation 36 reduces to the uniform distribution on the unit-sphere $f_U(u) = 1/\omega_{d-1}$.

Finally, it is possible to obtain a closed form in terms of a special function.

**Proposition B.7.** *With* $\lambda = (\boldsymbol{\mu}^T\Sigma^{-1}\boldsymbol{\mu})^{\frac{1}{2}}$ *and* $\gamma = \frac{\boldsymbol{u}^T\Sigma^{-1}\boldsymbol{\mu}}{(\boldsymbol{u}^T\Sigma^{-1}\boldsymbol{u})^{\frac{1}{2}}}$, *the probability density of the normalized Gaussian vector is*

$$
f_U(\boldsymbol{u}) = \frac{(\boldsymbol{u}^T\Sigma^{-1}\boldsymbol{u})^{-\frac{d}{2}}}{(2\pi)^{\frac{d}{2}}|\Sigma|^{\frac{1}{2}}w}\,e^{-\frac{1}{2}\lambda^2 - \frac{1}{8}\gamma^2}\Gamma(d)D_{-d}\left(\sqrt{2}\gamma\right), \tag{37}
$$

*where* $D_{-d}$ *is a Parabolic cylinder function.*

*Proof.* A result in the celebrated Tables of integrals, Series and Products of Gradshteyn and Ryzhik states, (Zwillinger et al., 2014, eq. 3.462), that

$$
\int_0^\infty x^{\nu-1}e^{-\beta x^2 - \gamma x}\mathrm{d}x = (2\beta)^{-\nu/2}\Gamma(\nu)e^{-\frac{\gamma^2}{8\beta}}D_{-\nu}\left(\frac{\gamma}{\sqrt{2\beta}}\right)\text{ for }\beta > 0, \nu > 0 \tag{38}
$$

where $D_\nu$ is a parabolic cylinder function, (Zwillinger et al., 2014, eq. 9.240). We see that the integral in equation 27 has precisely this form, with $\nu = d$, $\beta = 1/2$, and $\gamma = \lambda\bar{u}^T\Sigma^{-1}\bar{\boldsymbol{\mu}}$. Plugging this in equation 27 and rearranging yield equation 37. $\qquad\square$

**Corollary B.8.** *Let* $p, d \in \mathbb{N}^{+\star}$. *For* $\boldsymbol{z} \in \mathbb{R}^d$ *following a* $d$-*variate Gaussian of mean* $\boldsymbol{\mu} \in \mathcal{S}_p^d$ *and covariance matrix* $\Sigma = \sigma^2 I$, *the distribution of* $\boldsymbol{u}$, *the projection of* $\boldsymbol{z}$ *on* $\mathcal{S}_p^d$ *such that* $\boldsymbol{u} = T_{l_p}(\boldsymbol{z})$ *is defined by:*

$$
g_\kappa^{PGD}(\boldsymbol{u}, \boldsymbol{\mu_c}) = a_\kappa e^{-\frac{1}{2}\kappa^2}\sum_{n=0}^\infty \frac{(\kappa\frac{\boldsymbol{u}^T\cdot\boldsymbol{\mu}}{||\boldsymbol{u}||_2\cdot||\boldsymbol{\mu}||_2})^n\,\Gamma\left(\frac{d}{2}+\frac{n}{2}\right)}{n!\,\Gamma\left(\frac{d}{2}\right)} \tag{39}
$$

*with* $\kappa^2 = \frac{||\boldsymbol{\mu}||_2}{\sigma^2}$ *and* $a_\kappa$ *a normalization factor.*

*Proof.* Starting from equation 34 leads to equation 39 with $a_\kappa = \frac{\Gamma\left(\frac{d}{2}\right)(\boldsymbol{u}^T\boldsymbol{u})^{-\frac{d}{2}}}{2\pi^{\frac{d}{2}}w}$ $\qquad\square$

## C  PROOF OF PROPOSITION 3.2

Trivial starting from Equation equation 19 and replacing $r_c$ by $p_c$.

## D  HYPER-PARAMETER SEARCH

We conducted a small hyper-parameter for the optimizer and $v$ to obtain the results presented in Table 1. The values tested are presented in Table 2.

### D.1  HARDWARE AND COMPUTATION

For the compared methods, we trained on RTX A5000 for 300 epochs. The training time consumption is 4 hours for CIFAR10 and CIFAR100 and 60 hours for ImageNet100.

| Loss | Parameter | Values |
|---|---|---|
| | | CIFAR10 |
| SCE | optim | [SGD, Adam] |
| | lr | [0.0001, 0.001, 0.01, 0.1] |
| SCE-$\tau$ | optim | [SGD, Adam] |
| | lr | [0.0001, 0.001, 0.01, 0.1] |
| | $v$ | [0, 0.5, 1, 1.5, 2, 2.1, 2.2, $\cdots$, 3, 4] |
| SCE-$\tau$, $p=0.5$ | optim | [SGD, Adam] |
| | lr | [0.0001, 0.001, 0.01, 0.1] |
| | $v$ | [0.005, 0.006, 0.007, 0.008, 0.009, 0.01, 0.01, 0.1, 0.2, $\cdots$, 1.0] |
| SCE-$\tau$, $p=1$ | optim | [SGD, Adam] |
| | lr | [0.0001, 0.001, 0.01, 0.1] |
| | $v$ | [0.05, 0.1, 0.15, $\cdots$, 0.95, 1] |
| SCE-$\tau$, $p=1.5$ | optim | [SGD, Adam] |
| | lr | [0.0001, 0.001, 0.01, 0.1] |
| | $v$ | [0.05, 0.1, 0.15, $\cdots$, 0.95, 1] |
| SCE-$\tau$, $p=2$ | optim | [SGD, Adam] |
| | lr | [0.0001, 0.001, 0.01, 0.1] |
| | $v$ | [0.05, 0.1, 0.15, $\cdots$, 0.95, 1] |
| SCE-$\tau$, $p=3$ | optim | [SGD, Adam] |
| | lr | [0.0001, 0.001, 0.01, 0.1] |
| | $v$ | [0.05, 0.1, 0.15, $\cdots$, 0.95, 1] |
| SCE-$\tau$, $p=\infty$ | optim | [SGD, Adam] |
| | lr | [0.0001, 0.001, 0.01, 0.1] |
| | $v$ | [0.05, 0.1, 0.15, $\cdots$, 0.95, 1] |
| | | CIFAR100 |
| SCE | optim | [SGD, Adam] |
| | lr | [0.0001, 0.001, 0.01, 0.1] |
| SCE-$\tau$ | optim | [SGD, Adam] |
| | lr | [0.0001, 0.001, 0.01, 0.1] |
| | $v$ | [0, 0.5, 1, 1.5, 2, 2.1, 2.2 $\cdots$, 3, 4] |
| SCE-$\tau$, $p=0.5$ | optim | [SGD, Adam] |
| | lr | [0.0001, 0.001, 0.01, 0.1] |
| | $v$ | [$1e^{-5}$, $2e^{-5}$, $\cdots$, $1e^{-4}$, $1e^{-3}$, $1e^{-2}$, 0.1, 0.2, $\cdots$, 1.0] |
| SCE-$\tau$, $p=1$ | optim | [SGD, Adam] |
| | lr | [0.0001, 0.001, 0.01, 0.1] |
| | $v$ | [0.001, 0.002, $\cdots$, 0.01, 0.02, $\cdots$, 0.1, 0.2, $\cdots$, 1] |
| SCE-$\tau$, $p=1.5$ | optim | [SGD, Adam] |
| | lr | [0.0001, 0.001, 0.01, 0.1] |
| | $v$ | [0.005, 0.01, $\cdots$, 0.1, 0.2, $\cdots$, 0.1, 0.2, 1] |
| SCE-$\tau$, $p=2$ | optim | [SGD, Adam] |
| | lr | [0.0001, 0.001, 0.01, 0.1] |
| | $v$ | [0.01, 0.02, $\cdots$, 0.05, 0.1, 0.15, $\cdots$, 0.95, 1] |
| SCE-$\tau$, $p=3$ | optim | [SGD, Adam] |
| | lr | [0.0001, 0.001, 0.01, 0.1] |
| | $v$ | [0.01, 0.02, $\cdots$, 0.03 0.1, 0.2, $\cdots$, 1] |
| SCE-$\tau$, $p=\infty$ | optim | [SGD, Adam] |
| | lr | [0.0001, 0.001, 0.01, 0.1] |
| | $v$ | [0.05, 0.1, 0.15, 0.16, $\cdots$, 0.3, 0.4, $\cdots$, 1] |
| | | ImageNet100 |
| SCE | optim | [Adam] |
| | lr | [0.0001] |
| SCE-$\tau$ | optim | [Adam] |
| | lr | [0.0001] |
| | $v$ | [2.7] |
| SCE-$\tau$, $p=0.5$ | optim | [Adam] |
| | lr | [0.0001] |
| | $v$ | [$1e^{-5}$, $2e^{-5}$, $\cdots$, $1e^{-4}$, $1e^{-3}$, $1e^{-2}$, 0.1, 0.2, $\cdots$, 1.0] |
| SCE-$\tau$, $p=1$ | optim | [Adam] |
| | lr | [0.0001] |
| | $v$ | [0.007] |
| SCE-$\tau$, $p=1.5$ | optim | [Adam] |
| | lr | [0.0001] |
| | $v$ | [0.02, 0.025,0.030, 0.035] |
| SCE-$\tau$, $p=2$ | optim | [Adam] |
| | lr | [0.0001] |
| | $v$ | [0.05] |
| SCE-$\tau$, $p=3$ | optim | [Adam] |
| | lr | [0.0001] |
| | $v$ | [0.09] |
| SCE-$\tau$, $p=\infty$ | optim | [Adam] |
| | lr | [0.0001] |
| | $v$ | [0.12, 0.19, 0.2, 0.21, 0.22, 0.23] |

Table 2: Hyper-parameters for every method on CIFAR10, CIFAR100 and ImageNet100

