# OpenReview forum: "Learning representations on Lp hyperspheres: The equivalence of loss functions in a MAP approach"
_ICLR.cc/2025/Conference — ICLR 2025 Conference Withdrawn Submission_

### Official Review · Reviewer_z11Y · 2024-10-21

**Soundness:** 3
**Presentation:** 3
**Contribution:** 3
**Rating:** 5
**Confidence:** 3

**Summary:**

The paper provides a unified theoretical framework for learning representations on any Lp hypersphere in deep neural networks, extending the common practice of using L2 hyperspheres as the projection layer. The authors first show that SCE-τ and its variants can be interpreted through MAP modeling, connecting these loss functions to Gaussian assumptions. This work explained how the temperature parameter in SCE-τ is related to the variance of the Gaussian distribution. Second, the paper introduces PGD, which generalizes Gaussian distributions on Lp hyperspheres. Finally, the authors tried to demonstrate the theoretical equivalence of projections on different Lp hyperspheres within the MAP framework. They empirically validate that SCE-τ with optimal temperature values and PGD can achieve comparable performance.

**Strengths:**

- To my knowledge, I believe the author made a novel explanation of SCE losses and the connection w/ Lp projection is interesting.
- The quality of the paper is overall good and the writing is clear.
- I think this paper can have a significant impact because although SCE + L2-projection is widely used in many learning settings, including contrastive learning, etc., the theoretical reasonings behind are rarely analyzed rigorously. The author tried to build a MAP-based, a classic tool for analysis, framework to explain the mechanism of SCE-τ with Lp-projection. I can see this type of analysis can be possibly applied to more losses. With their framework, the proposed PGD loss works well without finding the optimal p. Besides, the authors provide some empirical evidence to validate their results.

**Weaknesses:**

- Insufficient comparison with related work: Although authors cited [1], due to the high similarity of [1]’s Section 3.1 and this paper’s Section 3.1+Section 3.2 (which is understandable because they both use MAP framework), I would recommend authors to elaborate on the difference and similarities between these two works. This will help readers to understand your contributions better.
- Additional experiments to verify claims: although I enjoy reading the analysis in the paper, extra experiments about several claims will strengthen your paper. See questions section.

I would love to raise my score if you can address all my concerns and questions below.


[1] Michel, N., G. Chierchia, R. Negrel, and J.-F. Bercher. “Learning Representations on the Unit Sphere: Investigating Angular Gaussian and Von Mises-Fisher Distributions for Online Continual Learning”. Proceedings of the AAAI Conference on Artificial Intelligence, vol. 38, no. 13, Mar. 2024, pp. 14350-8, doi:10.1609/aaai.v38i13.29348.

**Questions:**

- Line 206: “the Gaussian assumption on the projection on the hypersphere is questionable”: I understand in later sections you don’t assume Gaussian assumption on hypersphere (after projection). Typically, the distribution on hyperspheres is modeled by von Mises-Fisher distributions, which is very analogous to Gaussian distributions but not the same. Is there any reason why you pick the Gaussian assumptions in this paper?
- Line 219-220: Proposition 3.2, Add the reference of the proof in the Appendix.
- Line 233: “We showed that … considered Gaussian.” The proof of proposition 3.1 seems valid. Since Prop. 3.1 is quite important for your paper, I would love to see experiments to verify this claim. Something that you can try is to use synthetic Gaussian data to show the equivalence between two losses.
- Line 318-319: “Hence, … value of p.” Here for the wording “proportional”, do you mean the optimal v will change with p? Or will v grow with p? From Table 1 it seems you mean the latter, but proportional is a stronger statement where you mean v = some scalar factor * p. Maybe consider rewriting this sentence. Additionally, can you elaborate on why v will grow with p? From line 314-318, I can only interpret this sentence as the optimal v will change with p.
- Line 300-302: “In our setting, … small variance values.” How small should the variance value be for equivalence? From line 306-313, the validity of Gaussian assumption will be established when v -> 0, but in Table 1, the optimal v can be large (ex., CIFAR10) for reasonable performance. How to explain this discrepancy?
- Table 1: Can you use your framework to explain the performance boost from using projection, i.e., the difference between first two rows and others below?
- Figure 2, 4: why are some var values missing for some p? For example, in Figure 4, CIFAR10 values for var = 0.75-1.0 seem missing.
- Figure 2-3: Consider using the same color for the same p value.
- Figure 4: What causes the difference between the trend seen in two datasets? Is it because of the difficulty of the classification problem? If yes, why?
- Line 458-459: “when v is too large, … modeled Gaussian” Can you elaborate on this? Is this because v is the variance of the Gaussian and larger v will cause more overlap? Is there any relationship between the claim made in [1]’s Figure 2, which says “When classes are well-clustered (forming spherical caps), they are linearly separable”. As a follow up question, how do you evaluate the classification accuracy in Table 1? Is it applying a linear classifier on top of the frozen feature or something else?

[1] Wang, Tongzhou, and Phillip Isola. "Understanding contrastive representation learning through alignment and uniformity on the hypersphere." International conference on machine learning. PMLR, 2020.

---

### Official Review · Reviewer_bG4U · 2024-11-03

**Soundness:** 2
**Presentation:** 2
**Contribution:** 2
**Rating:** 5
**Confidence:** 3

**Summary:**

The paper established theoretical connections between the Softmax Cross-entropy (SCE), and its variants.  The paper concluded that SCE-τ can be interpreted as a MAP with a class-conditional isotropic Gaussian hypothesis on the standard simplex. Based on this link, the paper then introduced the Projected Gaussian Distribution (PGD) to model Gaussian distributions projected on any Lp hypersphere. The empirical evidence shows that by extending SCE to PGD and SCE-τ, the classification enjoys certain advantages over original SCE.

**Strengths:**

The paper tries to establish theoretical connections between the Softmax Cross-entropy (SCE), and its variants. Based on this link, the paper then introduced the Projected Gaussian Distribution (PGD) to model Gaussian distributions projected on any Lp hypersphere. The work is novel in the sense that it establishes the connection between SCE and its variants and then further extends it by leveraging the PGD.

**Weaknesses:**

W1. The paper lacks empirical results over larger dataset like Imagenet1K and on larger deep architectures such as ResNet50 or even transformers. The demonstrated results could be sensitive or prone to small dataset or small capacity of small architectures. I also hope to see the experiments on broader applications such as detections, or segmentations that all involve classifications.

W2. The theoretical support in the paper is weakly linked to the empirical result how and why the classification can be improved based on the PGD. Why unifying SCE with its variants and introducing PGD can improve the classification result? This remains unclear to me from the paper.


W3. Presentation, some section only has a single equation, without further instantiations of the equation, see Section 4.3. There is typo in section title 3.4.

**Questions:**

1. Why unifying SCE with its variants and introducing PGD can improve the classification result? What is the significance of the proposed theories in guaranteeing a better classification result.

2. Does the proposed method still show advantages on larger dataset like ImageNet1K and on larger architecture such as ResNet50/ViT-transformer?

---

### Official Review · Reviewer_YgVL · 2024-11-04

**Soundness:** 3
**Presentation:** 4
**Contribution:** 3
**Rating:** 6
**Confidence:** 3

**Summary:**

This paper establishes a connection between the MAP approach and SCE and proposes PGD loss to demonstrate their empirical insights into classification models. The theoretical analysis with the proofs provided offers an interpretation of SCE with temperature in terms of MAP. The evidence for this interpretation has been demonstrated through experiments on classification tasks. However, further investigation is required to ascertain the limitations of PGD loss, i.e., that it cannot be exploited when the number of classes is much larger than the dimension of projected features. Additionally, the effectiveness of PGD loss on practical applications still requires verification -- These points, which significantly affect the final decision of this review, will have to be addressed during the revision.

**Strengths:**

* The proposed method is well motivated based on proper related works and background.

* The interpretation of SCE with temperature in terms of MAP is a novel perspective on the loss function in classification models, and it has been well verified theoretically and experimentally.

* The proposed method benefits from stability regarding the temperature in SCE, which is a suboptimal problem in classification tasks.

**Weaknesses:**

**Weakness 1.** The primary concern pertains to the practical implications of PGD. As illustrated in Table 1, while PGD exhibits robust performance irrespective of $p$ value, its efficacy does not appear to surpass that of SCE-$\tau$. Could the authors elaborate on the distinctive advantages of PGD and the rationale behind its utilization?

**Weakness 2.** The confusion between a vector symbol and a random vector symbol often appears in the main paper and appendix (e.g., $\boldsymbol{z}$ and $\mathbf{z}$). As a result, this confusion undermines its overall confidence, despite the clear and detailed descriptions of the propositions and their proofs. The authors are kindly requested to examine and fix them thoroughly.

**Weakness 3.** SCE-$\tau$ with $p=2$ is arguably the most well-known method; however, the accuracy curve for different variances is not shown in Figure 3. Including the result in Figure 3 could provide a more compelling demonstration of the superiority of PGD loss.

**Weakness 4.** As illustrated in equations 11 and 12, the gradient flows in the classification models with PGD loss seem to be more complex. It would be beneficial to compare the computational costs, such as time and memory, between SCE-$\tau$ and PGD loss.

**Questions:**

**Question 1.** According to [1], for $K$ class weight vectors to form $d$-Orthoplex, the dimension of a latent representation should be larger than or equal to $\lceil \frac{K}{2} \rceil$. This constraint would be similarly considered in $L_{\infty}$ hypersphere to locate class distributions on each side of hypercube in isolation, and it might be the reason why there is no experimental results with Reset18 on ImageNet-1k where $d<\lceil \frac{K}{2} \rceil$ as $d=512$ and $K=1000$. The authors are kindly requested to address whether this kind of constraint really occurred or not. If it did not occur, why are there no results about ResNet18 with PGD loss on ImageNet-1k?

**Question 2.** Neural collapse [2] is a recently discovered phenomenon that when training error is 0, the last-layer features of the same class will collapse into a single vertex, and the vertices of all classes will be aligned with their classifier prototypes and be formed as a simplex equiangular tight frame. Could the authors compare PGD on $L_{p}$ hypersphere with the neural collapse? Similarly, it would be of interest to ascertain whether the interpretation of SCE-$\tau$ in terms of MAP covers the neural collapse or conflicts with it.

**Question 3.** How about making $v$ be a trainable parameter? If it is effective, could the suboptimal problem of temperature in SCE be alleviated?

---

**Things to improve the paper that did not impact the score:**

* [line 50-51] [3] has theoretically analyzed the feature normalization in classification tasks, based on an unconstrained feature model, and accounted for the effect of feature normalization with SCE loss like the authors but in terms of neural collapse [2], not MAP. Therefore, the paper would be enriched if [3] was covered in the introduction or related work.

* [line 116] Missing whitespace between $z_{c}$ and *the*

* [line 123] Duplicated *Michel et al.*

* [line 282] Where is $\mathbf{\mu}_{c}$ on the right-hand side? Or, is subscript $c$ of the left-hand side unnecessary?

* [line 331] Where is $z_{a}$? In my opinion, the first $z$ would be $z_a$.

* [line 392] Missing *equation* in front of *12*

* For the future work, it would be beneficial to utilize a fixed classifier [1, 4] and the neural collapse phenomenon [2].

---

[1] Pernici, F., Bruni, M., Baecchi, C., & Del Bimbo, A. (2021). Regular polytope networks. _IEEE Transactions on Neural Networks and Learning Systems_, _33_(9), 4373-4387.

[2] Papyan, V., Han, X. Y., & Donoho, D. L. (2020). Prevalence of neural collapse during the terminal phase of deep learning training. _Proceedings of the National Academy of Sciences_, _117_(40), 24652-24663.

[3] Yaras, C., Wang, P., Zhu, Z., Balzano, L., & Qu, Q. (2022). Neural collapse with normalized features: A geometric analysis over the riemannian manifold. _Advances in neural information processing systems_, _35_, 11547-11560.

[4] Hoffer, E., Hubara, I., & Soudry, D. (2018). Fix your classifier: the marginal value of training the last weight layer. *ICLR 2018*.

---

### Official Review · Reviewer_n7MX · 2024-11-05

**Soundness:** 2
**Presentation:** 2
**Contribution:** 2
**Rating:** 3
**Confidence:** 4

**Summary:**

This paper proposes a framework that demonstrates the equivalence of all projections on $L_p$ ball through Maximum A Posteriori (MAP) modeling, under the assumption that the network outputs can be approximated by a Gaussian distribution.

**Strengths:**

This paper proposes a framework that demonstrates the equivalence of all projections on $L_p$ ball through Maximum A Posteriori (MAP) modeling, under the assumption that the network outputs can be approximated by a Gaussian distribution. The idea is interesting.

**Weaknesses:**

Unfortunately, I also have many concerns about this paper.

The most fundamental concern is that it is unreasonable that the network output can be approximated as a Gaussian distribution. What does the approximation here specifically refer to, and on what distance is the approximation?

Other comments are listed below:

1. In proposition 3.1, the conditional probability density functions are assumed to be a isotropic Gaussian distribution. This assumption is used in the following content, and in the proof of proposition 3.1. Thus, the assumption should also be taken as an assumption of this article, not just the assumption that the network output is  approximated as a Gaussian distribution. The rationality of this assumption should also be discussed.

2. In proposition 3.2, I do not understand why the variance $v$ of the isotropic Gaussians is equal to one. It means that conditional probability density functions all possess the same variance matrix as identity matrix $\mathbb{I}$? The differences of conditional probability density functions are just the means of Gaussian distributions. This assumption seems not reasonable.

3. In Sec. 4.4, the authors assume $v$ tends to $0$ when they show that the ardial and axial projections tend to result in the same projections. I am afraid that this contradicts the above assumption, in proposition 3.2, that the variance $v$ of the isotropic Gaussians is equal to one. How could one tend to $0$?

4. This article is poorly organized. Figure 1 does not contain much information. These two figures (a) (b) are more like conducting mathematical popularization. I don't know what the meaning of Figure 1 is.

5. The formats of the references are inconsistent.

6. The overall experimental results cannot verify the theory proposed in this article. The accuracy results in the experiment cannot reflect the consistency of the different loss functions. I recommend to add more visualization results, such as the visualization of the gradient of the loss function, to further verify the equivalence between different loss functions.

**Questions:**

Please refer to the weaknesses above.

---

### Note · Authors · 2024-11-23

I have read and agree with the venue's withdrawal policy on behalf of myself and my co-authors.